# Transportation Mode Detection Combining CNN and Vision Transformer with Sensors Recalibration Using Smartphone Built-In Sensors

**DOI:** 10.3390/s22176453

**Published:** 2022-08-26

**Authors:** Ye Tian, Dulmini Hettiarachchi, Shunsuke Kamijo

**Affiliations:** 1Graduate School of Interdisciplinary Information Studies (GSII), The University of Tokyo, 4 Chome-6-1 Komaba, Meguro City, Tokyo 153-0041, Japan; 2The Institute of Industrial Science (IIS), The University of Tokyo, 4 Chome-6-1 Komaba, Meguro City, Tokyo 153-0041, Japan

**Keywords:** transportation mode detection, spectrogram recognition, sensors recalibration, CNN, vision transformer, lifelog

## Abstract

Transportation Mode Detection (TMD) is an important task for the Intelligent Transportation System (ITS) and Lifelog. TMD, using smartphone built-in sensors, can be a low-cost and effective solution. In recent years, many studies have focused on TMD, yet they support a limited number of modes and do not consider similar transportation modes and holding places, limiting further applications. In this paper, we propose a new network framework to realize TMD, which combines structural and spatial interaction features, and considers the weights of multiple sensors’ contributions, enabling the recognition of eight transportation modes with four similar transportation modes and four holding places. First, raw data is segmented and transformed into a spectrum image and then ResNet and Vision Transformers (Vit) are used to extract structural and spatial interaction features, respectively. To consider the contribution of different sensors, the weights of each sensor are recalibrated using an ECA module. Finally, Multi-Layer Perceptron (MLP) is introduced to fuse these two different kinds of features. The performance of the proposed method is evaluated on the public Sussex-Huawei Locomotion-Transportation (SHL) dataset, and is found to outperform the baselines by at least 10%.

## 1. Introduction

Recent developments in human activity recognition (HCR) and TMD have been applied in several fields, including ITS [1], context-aware positioning [2], health monitoring [3], and Lifelog [4].

Lifelog is a typical application of TMD; the users can check the record of Lifelog to recall historical events and carry out self-planning. Developers can leverage the data to personalize marketing or city development. Providing highly accurate and complicated TMD can contribute to increasing the richness of Lifelog semantics. For basic application scenarios, transportation modes are often simply divided into 5 groups: still, walking, running, in-road-vehicle, and in-rail-train. The place where the sensor is held on the body is usually not considered. In this study, we focus on realizing a higher level of TMD performance for 8 groups: still, walking, running, cycling, car, bus, railway, and subway [5], to provide higher-level semantic information for Lifelog, where ‘car and bus’ and ‘railway and subway’ are two groups of similar transportation modes that are subdivided from the two modes, in-road-vehicle and in-rail-train. At the same time, different holding modes of sensors are considered; these make the TMD task more difficult.

For recognizing the complicated transportation modes with different holding modes, we redefine the time sequence data-based TMD task as a spectrogram recognition task and propose a deep learning framework for TMD using features of time-frequency responses obtained via smartphone built-in sensor data. Our proposed framework comprises three key sub-modules including the structure feature extraction module, spatial interaction feature extraction module, and sensors weights recalibration module. We combine the recalibrated structure features and spatial interaction features to realize the recognition of the transportation modes.

The structure of this paper is as follows: Section 2 reviews state-of-the-art related works for transportation mode detection using smartphone built-in sensors based on machine learning or deep learning methods. Section 3 introduces the methodology and motivation of the proposed method. Section 4 describes the experimental setup and dataset, and gives the analysis of evaluation results. Finally, Section 5 concludes with a summary and future work.

## 2. Related Work

The mainstream methods of TMD include trajectory-based methods [6] and sensors-based methods [7]. Among the two, the sensor-based method is characterized as low-cost and real-time due to the limitations of GPS in urban areas.

### 2.1. Machine Learning-Based Methods

Over the last decades, many researchers have contributed to transportation mode detection based on smartphone built-in sensors. Most of these works have adopted machine learning frameworks including the Support Vector Machine (SVM) [8], Decision Tree (DT), and Random Forest (RF) [9,10] to realize TMD. Ashqar et al. proposed a Hierarchical Machine Learning Classifier (HMLC) method [11], which includes a two-layer hierarchy framework which uses manually-extracted time-domain and frequency-domain features. In other studies, Li Dailin et al. proposed a method based on the Hidden Markov Model (HMM) and RF [12], the STD-Means of raw data is processed as an input, and Janko et al. proposed a JSI-Classic method [13] which, combined with five machine learning sub-modules, including RF, DT, gradient boosting (GB), SVM, and K-nearest neighbors (KNN), used manually-extracted features as input and fusion the outputs of each sub-module by ensemble learning. None of these methods recognize similar transportation modes and multiple holding modes, but with the development of optimization methods, the deep learning methods can be combined with the optimization methods [14], which will provide a new solution to TMD.

### 2.2. Deep Learning-Based Methods

With the recent development of deep learning, many works have proposed TMD methods based on the deep learning framework including CNN [15,16] and LSTM [17]. In other advanced studies, Chen et al. proposed the ABLSTM [18] method combining BLSTM and attention mechanisms. Saeed et al. proposed a self-supervised deep network, named the Transformation Prediction Network (TPN) [19], to recognize transportation modes, and is composed of several convolutional layers, using self-supervised pre-training to improve the recognition performance. The above methods are only effective when recognizing basic transportation modes and single holding modes. Some studies have tried to provide solutions for similar transportation modes: Gjoresk et al. proposed an optimized JSI-CLASSIC method [20] which firstly constructed some sub-learners including RF, gradient boosting, SVM, AdaBoosting, KNN, naïve Bayes, DT, and Deep Neural Network (DNN), and then coupled them to generate the transportation mode prediction. Ordóñez et al. present a Deep-Conv LSTM framework [21] composed of convolutional layers and an LSTM layer, this inputs the time domain data of multiple sensors into the convolutional layer without pre-processing, and further processes the output of the convolutional layer by the LSTM layer. C. Zhang et al. use CNN to learn features of Signal images, Gramian Angular Field images, Markov Transition Field images, and Recurrence Plot images and then combine the four kinds of features using SVM [22]. A. Sharma et al. use CNN and LSTM [23] to learn the time correlation, and a decision policy is defined on top of the classifier to perform the transportation mode prediction for the incoming time series by attaining an acceptable trade-off. Wang et al. proposed the MSRLSTM [24], which introduced the residual CNN and attention module into Deep-Conv LSTM to further improve the performance. These methods can recognize similar transportation modes but are not effective for multiple holding modes.

Unlike the aforementioned methods, our work aims to address the challenges in recognizing similar transportation modes and multiple holding modes. The main contributions of this paper are summarized as follows: (1) we use the sliding window mechanism to segment the raw data and transform it into time-frequency spectrograms using continuous wavelet transforms (CWTs) to learn higher-level features. (2) For the spectrogram recognition task, we propose a framework composed of three sub-modules including ECAnet [25], Resnet [26], and Vit [27] using accelerometer, gyroscope, and magnetometer data. (3) We discuss the effect of the magnetometer for TMD and provide the motivation and evaluation of each sub-module. We evaluate our algorithm on a large public SHL dataset [5], and compare with baselines including LSTM [17], ABLSTM [18], TPN [19], Deep-Conv LSTM [21], and MSRLSTM [24]. This demonstrates that our proposed method performs superior to these baselines in the SHL dataset.

## 3. Motivation and Methodology

The overall framework of our proposed algorithm is given in Figure 1. The framework accepts sensor time sequence data and pre-processes using sliding window and CWT before extracting structure and interaction features. Features are calibrated with sensor weights to predict the transportation mode. The motivation and technical details of each sub-module are explained in the following sections.

We followed the transportation mode definitions in the SHL dataset [5], definitions are provided in Table 1. ‘Still’ was defined as ‘still without vehicles’, if the ‘Walk’, ‘Run’, and ‘Bike’ were stopped at the midway, the transportation modes were changed to ‘Still’. If ‘Car’, ‘Bus’, ‘Railway’, and ‘Subway’ were stopped at the midway, the transportation mode remained unchanged, as passengers did not get off. To avoid ambiguity caused by terms used in different regions, the ‘Railway’ was defined as a long-distance train which runs between and corresponds to ‘Train’ in the SHL dataset, and the ‘Subway’ was defined as a short-distance metro which runs within a city.

### 3.1. Data Input

We used the raw time sequence data of smartphone built-in sensors collected in the real world as input. Although using more sensors can achieve higher accuracy [28], considering the different hardware configurations of smartphones (some data such as linear acceleration and barometer are not permanently available, the computational cost of the network, etc.) we did not choose linear acceleration or barometer, and processed the raw triaxial data of accelerometer, gyroscope, and magnetometer into modulus as input.

### 3.2. Data Pre-Processing

As shown in Figure 2, we firstly segmented the input data by using a short-term sliding window with length of several seconds, and moved forward by a stride. After segmentation, we transformed each segmented data into a time-frequency domain. This process was applied to sequence data of the selected three sensors (accelerometer, gyroscope, magnetometer) to obtain a three-channel spectrogram. Preprocessing steps are explained in the following sections. The detail of input data and parameter of pre-processing are explained in the Section 4.

#### 3.2.1. Motivation of Data Segmentation

We proposed using a sliding window mechanism to segment the sensor time sequence data. For transportation mode, the information in short-term temporal dependencies of sensor raw data was much larger than that in the long-term temporal dependencies. The common transportation modes are periodic: walking and running are composed of periodic leg-lift and leg-fall, and subways and railways periodically pass through the gaps on the rails. ‘Still’ is an action with an infinite period. We only needed to capture more than one period (about 1 or 2 s) of information to detect the transportation mode. Long-term temporal dependencies did not help our task, and rather had a negative impact. When the time span is lengthened, the transportation mode in the k min had little relation to the transportation modes in the k + 1 min, which depends entirely on the user’s random decision.

#### 3.2.2. Motivation of Time-Frequency Transformation

The noise of real-world dataset composition includes not only stationary noise such as bias and measurement but also time-varying and non-stationary noise caused by different holding modes of the sensor (such as bag, hips, and torso) and random shaking of the human body (e.g., swinging arms). Due to the superposition of these noises, the transportation mode dataset collected from the real world can be regarded as a time-varying and nonstationary signal. Figure 3 shows example segments of different transportation modes and holding positions from the SHL dataset. In the time domain, it is difficult to distinguish transportation modes using the LSTM-based method due to the overlapping of the above noises. This is proved by the experiment result of baselines given in Section 4.

The CWT is commonly used in the time-varying and nonstationary signal processing issues [29], it allows us to analyze the characteristics of time-sequential data in time-frequency domains with high-resolution features. We adopted the CWT to process the segmented data of real-world transportation detection. As shown in Figure 4, the transformed data is 2D time spectrum images, where *x*-axis is time, *y*-axis is frequency, and the value of each pixel is time-frequency response intensity. We used the shuffled spectrum images for training, allowing the network to learn short-term temporal dependence and avoid the impact of long-term temporal dependence.

### 3.3. Multiple Sensors Integration and Recalibration

Multi-sensor combination will provide more information to the neural network and the TMD accuracy will increase by increasing the number of sensors used [28]. In addition to using inertial sensors including accelerometers and gyroscopes to measure human movements directly, we also used magnetometer data as an additional input. We took into consideration the fact that cars, buses, railways, and subways are all metal vehicles and so the surrounding geomagnetic field will be distorted because of the metal. We leveraged the distortion as extra information. In order to cater to a variety of smartphone built-in sensor platforms, we considered the hardware configurations and did not use sensors such as barometer, direction measurement, and linear accelerometer (no gravity). In order to reduce the calculation, we processed the modulus of triaxial data of each sensor (accelerometer, gyroscope, and magnetometer) into a spectrum image and considered each sensor as a channel to construct a three-channel image sensor as an input to the network.

Different sensors contribute differently to different transportation modes, for example, magnetometers should have a greater contribution to recognizing the modes related to metal vehicles, but would have a smaller contribution when recognizing non-metal modes such as running and walking. Figure 5 shows example segments from the SHL dataset with magnetometer distortion. If a metal vehicle passes by a pedestrian, the data from the accelerometer worn on the pedestrian remains stationary, but the magnetometer data is seriously distorted. Such magnetic distortion is fault information for non-vehicle modes.

To better describe the contribution of each sensor, we inserted the ECA-net into each convolutional layer. As shown in Figure 6, the ECA-net is a lightweight module, which uses Global Average Pooling (GAP) to pool the information of each channel. The data *X* with size height (*h*) * width (*w*) * channel (*c*) is input into ECA-net in the direction of the arrows. ECA-net converts the multi-channel picture tensor with *c* channels into a vector with a length of *c*, then traverses the vector with a 1D convolution kernel to learn the interactive information of adjacent channels, through a sigmoid active layer *σ* then multiplies it as a weight with the output of the previous layer to realize the weighting of multi-channel information. Finally, the data $\bar{X}$ learning channel attention is output to the next layer.

### 3.4. Feature Extraction and Fusion

To better express transformed data, we extracted the structure features of the spectrograms, and considered the spatial position interaction of frequency response in the spectrograms at the same time.

In spectrogram recognition, the most basic feature is the intensity distribution of time-frequency response in the time-domain and frequency domain, which is embodied in the fine-grained structure feature on the spectrogram. We adopted Resnet to extract this structure feature. As shown in Figure 7, Resnet is a classic CNN-based method, which uses the residual block that connects the convolution layer and residual propagation in parallel. It allows the convolution network to have deeper layers, so that higher-dimensional features in the image can be extracted. The data *X* is input into the residual block in the direction of the arrow and the $\bar{X}$ is output to the next layer. Where × means multiplication operation of channel attention and input data, + means the multilayer convoluted data is added with the single convolutional layer convolution shortcutting data to realize a residual block.

Differently from conventional image recognition tasks, for spectrum images, the *x*-axis is time and the *y*-axis is frequency. Besides the shape and strength of frequency response, the position distribution of frequency response in different frequency bands (i.e., the position of frequency response in the spectrum diagram) and the spatial interaction of each frequency response should not be neglected. However, due to the limitation of convolutional networks, convolution-based methods cannot learn the position and spatial interaction information of frequency response in the images [30]. We used Vit [27] to learn this space interaction information. Vit’s architecture is completely different from CNN’s; as shown in Figure 8, it divides the original input image into patches, and adds a specific position code to each patch to distinguish the position of the patch in the original picture. It then calculates the multi-head self-attention scores between each patch and all other patches. Through this process, the network will learn the spatial interaction information of the image.

Vit flattens each patch into a sequence as the input of the network, leading to loss of fine-grained structure information of the image, which is complementary to the structure information learned by Resnet.

## 4. Experiments and Analysis

In this section, we evaluate the performance of the proposed methods on the 2020 Sussex-Huawei Locomotion-Transportation (SHL) Dataset. We firstly introduce the detail of the dataset and experimental setup and discuss the effect of each sub-module on the performance of the proposed method. Lastly, we compare the performance with other state-of-the-art baselines.

### 4.1. Dataset

The public SHL dataset contains eight transportation modes: Still (127 h), Walk (127 h), Run (21 h), Bike (79 h), Car (88 h), Bus (107 h), Railway (115 h), and Subway (89 h). The data were collected by four users with four smartphones (Model: HUAWEI Mate 9). These smartphones were separately placed on four locations including Hands, Torso, Hips, and Bag. The dataset was collected in multiple rounds over 7 months, and the smartphone was randomly placed in hip pocket (right or left) and arm (right or left) during each collection. The sampling frequency of sensors is 100 Hz, and each sample includes triaxial accelerometer, triaxial gyroscope, triaxial magnetometer, triaxial linear acceleration, and scalar barometer.

### 4.2. Preprocessing

The data was processed into time-frequency spectrograms by using sliding window as mentioned in the Data Segmentation section; the parameters of preprocessing are listed in Table 2. In order to ensure the sample balance, we randomly selected 30,000 spectrograms for each sensor holding place and each transportation mode for training and testing, about 267 h in total. In all experimental results, each transportation mode label contained samples of all four holding modes. The preprocessed data were separated into two parts: training part (80%) and testing part (20%).

### 4.3. Experimental Setup

The hardware and software environment are listed in Table 3.

All the experiments were conducted on the same platform. In this paper, we used the label accuracy, F1-score, and total accuracy to evaluate the proposed method and baselines. These metrics are defined as Equations (1)–(4):(1)$$Total\;Acuuracy=\frac{\sum_{i=1}^{K}TP_{i}}{N}$$
(2)$$Label\;{Accuracy}_{i}=\frac{TP_{i}}{TP_{i}+FP_{i}}$$
(3)Recalli=TPiTPi+FNi
(4)F1−scorei=2∗Recalli∗Label AccuracyRecalli+Label Accuracyi
where the *TP_i_* is the number of correctly classified samples of transportation mode *i*, *FP_i_* is the number of samples that incorrectly classified transportation mode *i* as other modes, *FN_i_* is the number of samples that incorrectly classified other transportation modes as mode *i*, *K* is the number of transportation modes and *N* is the total number of all experimental samples.

### 4.4. Recognition Performance of Sub-Module Integrations

In this section, we will illustrate the effect of each sub-module and magnetometer. First, we analyzed the effect of each sub-module on the total accuracy separately. The parameter detail is given in Table 4.

The total accuracy of each integration of sub-modules is shown in Figure 9, which shows that the proposed method achieved the best performance. The total accuracy of R50 (No CWT, No MAGN,), R50 (No MAGN), R50, R50 + ECA-net, Vit, and R50 + Vit + ECA-net are 74.70%, 80.03%, 84.30%, 91.28%, 81.01%, and 93.03%, respectively. We observed that the integration of more sub-modules can further improve the accuracy of transportation mode recognition.

We analyzed the effect of the magnetometer information and different sub-modules on recognizing different transportation modes in detail. Firstly, we discuss the performance of structure feature extraction under the framework, and prove the effect of adding CWT, magnetometer, and sensors weight. We compare the performance of using raw data as input and using CWT. When using the same sensor and the same model, the recognition performance after adding CWT was greatly improved. As shown in Table 5, compared to the original ‘R50 (No MAGN)’, the total accuracy of ‘CWT + R50 (No MAGN)’ was improved by more than 5%, and the F1-score of each mode was improved, but the recognition performance of ‘Run’ was not improved, even decreasing by about 0.4%. This shows that ‘Run’ is a mode that can be easily recognized in the time domain, and as the recognition performance using time-domain data is already close to 100%, using time-frequency domain features is not effective for recognizing ‘Run’. However, for other modes, using CWT was effective. We will then discuss the effect of adding magnetometer and sensors weights. Under the same weight of each sensor, as shown in ‘CWT + R50′ of Table 5, compared to ‘CWT + R50 (No MAGN)’, it can be observed that the F1-score of transportation modes with metal vehicles such as ‘Bus’, ‘Car’, ‘Railway’ and ‘Subway’ were improved after adding magnetometer data, but the F1-score of ‘Walk’, ‘Run’ and ‘Bike’ decreased instead. As mentioned in Section 3, this is due to the false information introduced from extra geomagnetic distortion from the environment when the transportation mode was irrelevant to metal. Therefore, we insert ECA-net module into R50, to recalibrate weights for each sensor. As shown in the ‘CWT + R50 + ECA-net’ of Table 5, the F1-score of each transportation mode is obviously improved when the network learns the contributions of different sensors.

Secondly, we will discuss the performance of spatial interaction feature extraction. As shown in Table 5, the performance of recognizing each transportation mode was lower than using structural features. There are two reasons for this [30]:Each patch would be flattened into a vector as input in Vit and the network will lose its structure information because of the flattening operation;Vit is eager for datasets. Due to the flattening operation mentioned above, it is difficult for Vit to extract the local information of each patch when the dataset is insufficient. If a Vit model is trained alone, the accuracy will exceed the CNN-based model when the dataset is more than 100 M, but our dataset size is only about 1 M.

Finally, we will discuss the performance of the completed proposed method. We introduced MLP to combine the structure and spatial interaction feature which are extracted by Resnet and Vit, respectively. It can be observed in Table 5 that compared with methods using structure features or spatial features separately, the recognition performance of all the eight transportation modes was improved in the SHL dataset. Even if the dataset is small, Vit will keep the focus on the global spatial interaction information [30], which can be combined with structure information to achieve higher accuracy. It can be observed that the F1-score for Bus, Car, Railway, and Subway, two groups of similar transportation modes, the recognition performance was higher than 85%, especially for Bus and Car, where the recognition performances was higher than 94%.

### 4.5. Comparison with Baselines

We evaluated our method with the baselines mentioned in Section 2. All the baselines were reimplemented using Pytorch platform according to the opensource code and the input raw data in the time domain which includes the accelerometer, gyroscope, and magnetometer. The parameter detail of each baseline is listed in Table 6.

The total accuracy of the proposed method and baselines are illustrated in Figure 10, and the label accuracy, recall, and F1-score are listed in Table 7. We can observe that the proposed method significantly outperforms the baselines on the SHL dataset. The total accuracy of LSTM, ABLSTM, Deep-ConvLSTM, TPN, and MSRLSTM are 53.21%, 58.42%, 81.30%, 77.72%, and 83.39%, respectively, and our proposed method achieved 93.03%, outperforming baseline methods by at least 10%.

As Section 3 mentioned, due to the time-varying and non-stationary noise caused by different holding modes of the sensor (such as bag, hips, and torso) and random shaking of the human body, it is difficult to learn the time-domain feature of raw data directly, and the recognition performance of LSTM was the lowest among all methods. ABLSTM and Deep-Conv LSTM introduce the attention mechanism in the time domain, and additional convolution operation basis on LSTM, respectively. These improved the recognition performance. TPN leveraged the CNN in the time domain, which did not obtain enough receptive fields with shallow layers; the performance was higher than the LSTM-only method, but was lower than Deep-Conv LSTM, which combines CNN and LSTM. MSRLSTM combined the CNN, LSTM and attention in the time domain, achieving the highest performance in the baseline methods, but it also only focused on the features of the time domain. The experiments above prove that time domain convolution and attention mechanisms have effects on time-varying and nonstationary overlapping noise to a certain extent; the recognition performance of dissimilar transportation modes with four holding modes was relatively high, but with four holding modes, the performance in distinguishing similar transportation modes was not good enough. Compared with baseline methods, the proposed method uses CWT, which fundamentally avoids the problems of time-varying noise and time-domain overlapping. According to the characteristics of the time-frequency spectrogram, the intensity and location distribution of frequency response are combined, which makes the proposed method achieve higher recognition performance for all eight transportation modes. In particular for ‘Car’ and ‘Bus’ and ‘Railway’ and ‘Subway’, the two groups of similar transportation modes, the F1-score of recognition performance was more than 8% (‘Car’), 14% (‘Bus’), and 18% (‘Railway’ and ‘Subway’), higher than all the baselines.

## 5. Conclusions

In this paper, a new deep learning framework for transportation mode detection is proposed. The proposed method transforms the pattern recognition problem of sequence data into a recognition problem of the spectrogram by transforming the basic smartphone built-in sensor data into time-frequency domain for feature extraction. Differently from traditional image recognition that uses only structural features, we consider the characteristics of the spectrogram, the position (spatial interaction feature) and intensity (structure feature) of frequency response. The framework is realized by combining structure and spatial interaction features extracted using CNN and Vit, respectively. Furthermore, a channel attention module is introduced to recalibrate the sensor weights, as different sensors contribute differently in recognizing different transportation modes. Compared with other baselines, the proposed method achieves the highest performance with four holding modes, especially for similar transportation modes (‘Bus’ and ‘Car’, ‘Railway’ and ‘Subway’). The proposed method outperforms baselines by at least 10% in total accuracy, and about 8% in ‘Car’, 14% in ‘Bus’, and 16% in ‘Railway’ and ‘Subway’.

In this study, only data from four users was used for training; generalization for multiple users is not considered. A further study could consider the multiple attributes of different users based on advanced optimization, which could contribute to increasing the generalization of TMD models.

## Figures and Tables

**Figure 1 sensors-22-06453-f001:**
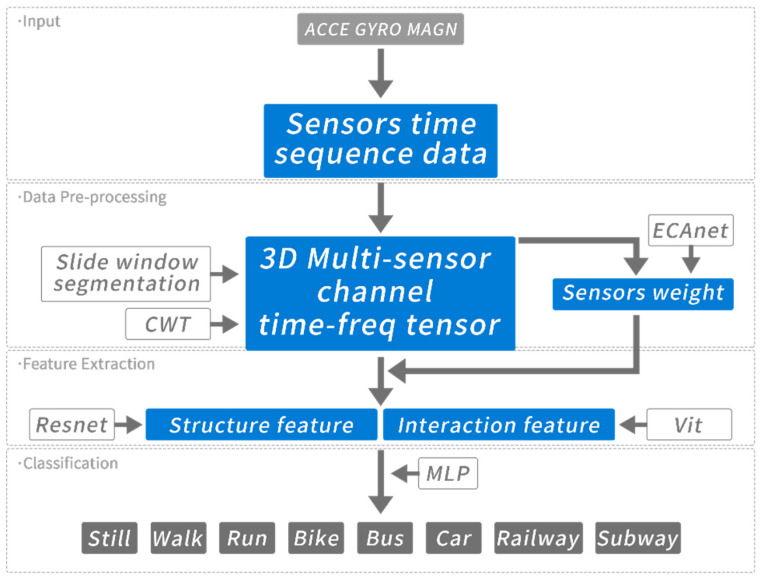
Overview of the proposed method.

**Figure 2 sensors-22-06453-f002:**
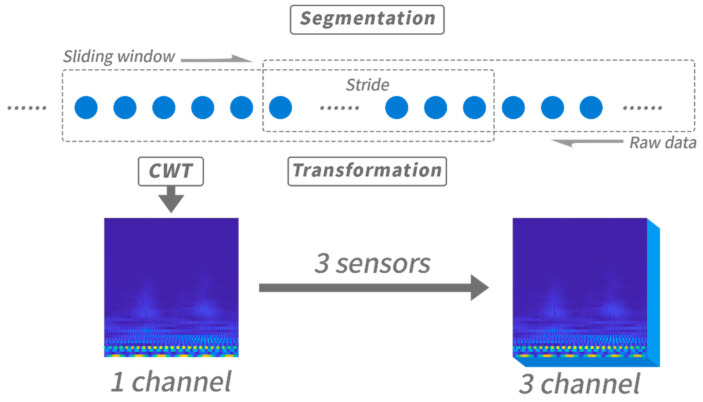
The overview of data pre-processing.

**Figure 3 sensors-22-06453-f003:**
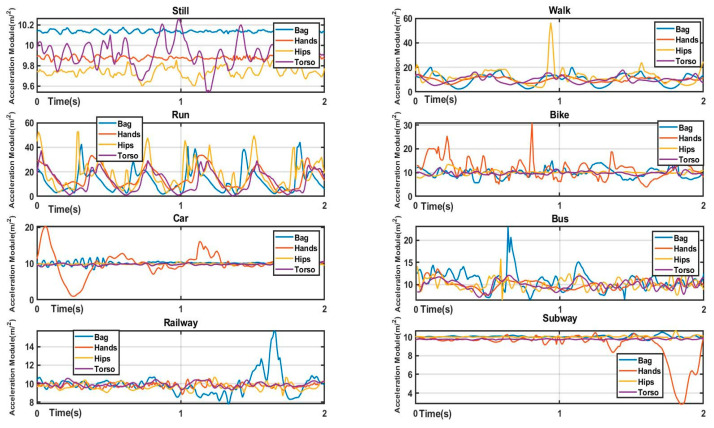
Raw acceleration data of eight transportation modes and four holding modes in SHL dataset in a sliding window.

**Figure 4 sensors-22-06453-f004:**
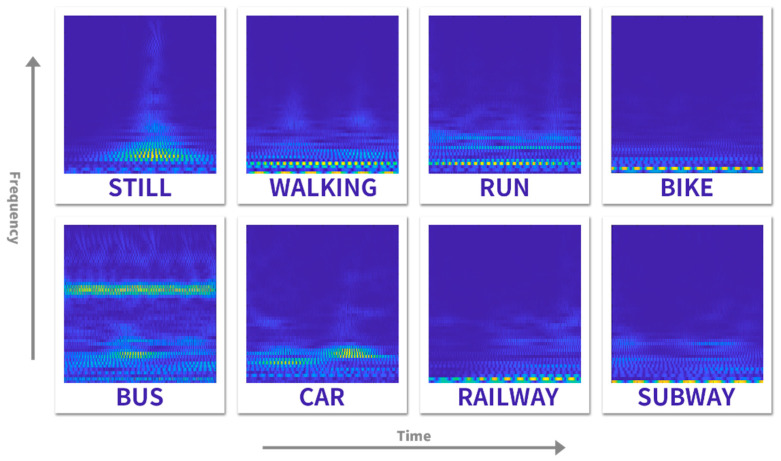
The processed time-frequency spectrogram.

**Figure 5 sensors-22-06453-f005:**
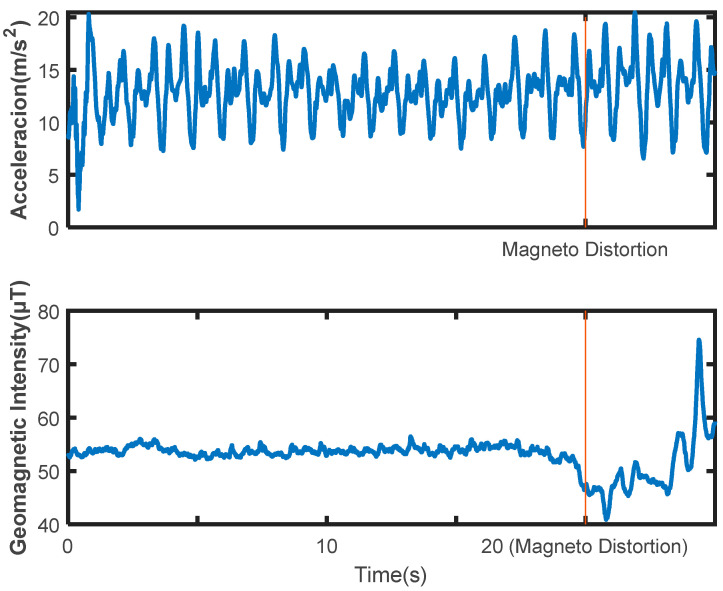
Geomagnetic distortion data segment with holding mode of ‘Hands’ in the SHL dataset.

**Figure 6 sensors-22-06453-f006:**
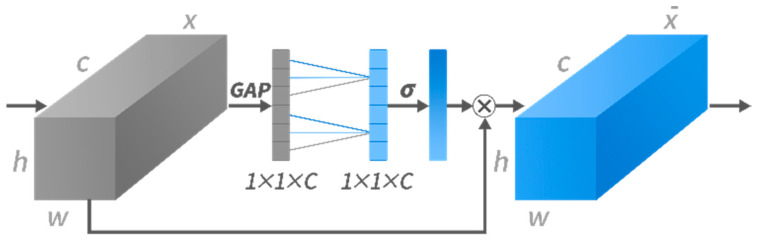
The architecture of ECA-net in a convolution layer.

**Figure 7 sensors-22-06453-f007:**
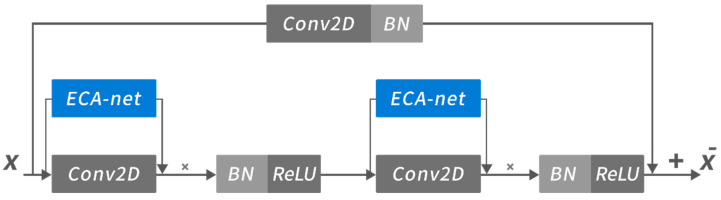
The architecture of ECA-net in a residual block.

**Figure 8 sensors-22-06453-f008:**
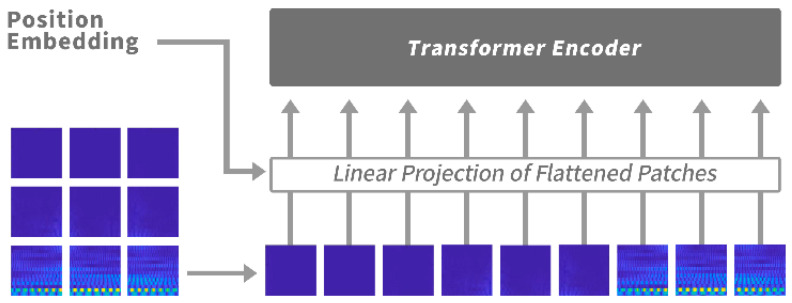
The architecture of Vit module.

**Figure 9 sensors-22-06453-f009:**
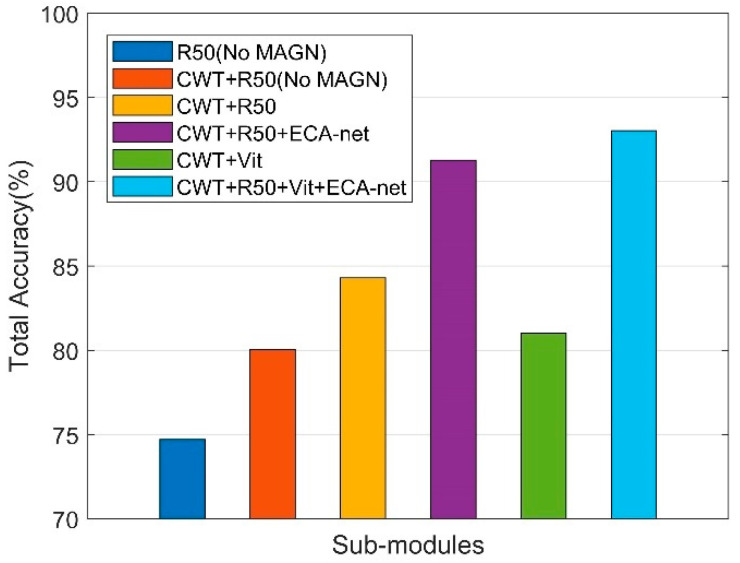
Total accuracy of different combination of sub-modules.

**Figure 10 sensors-22-06453-f010:**
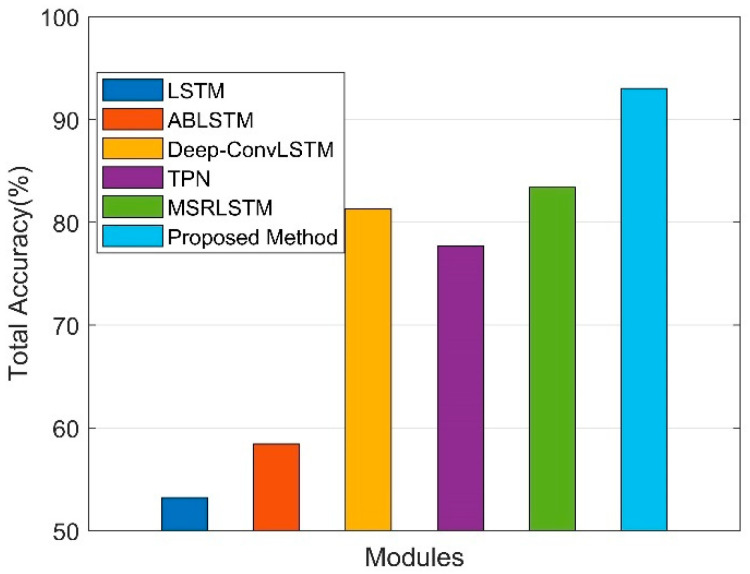
Total accuracy of the proposed method and other baselines.

**Table 1 sensors-22-06453-t001:** Transportation mode definition.

Transportation Mode	Definition
Still	Still in the open area
Walk	Walking in the open area
Run	Running in the open area
Bike	Cycling in the open area
Car	Still and waiting for traffic lights or running
Bus	Still at the station or running
Railway	Still at the station or running
Subway	Still at the station or running

**Table 2 sensors-22-06453-t002:** Parameter setting of preprocessing.

Parameter Name	Parameter Setting
Sliding window length	3 s
Sliding stride length	1 s
Wavelet transform scale	64
Wavelet name	Cmor3-3

**Table 3 sensors-22-06453-t003:** Environment of experiment.

Hardware and Software Environment	
CPU	AMD Ryzen 7 5800X 8-Core
Memory	16 GB
GPU	GTX 3060Ti 8GB x1
Development Language	Python 3.9
Framework	Pytorch 1.9

**Table 4 sensors-22-06453-t004:** Parameter setting of the proposed method.

Parameter Setting	Resnet	Vit	ECA-Net	MLP
Conv Layer 1	CNN(3),144	/	/	/
Conv Layer 2	$\left[\begin{array}{c} CNN(1),144\\ CNN(3),144\\ CNN(1),256 \end{array}\right]\times 3$	/	/	/
Conv Layer 3	$\left[\begin{array}{c} CNN(1),128\\ CNN(3),128\\ CNN(1),512 \end{array}\right]\times 4$	/	/	/
Conv Layer 4	$\left[\begin{array}{c} CNN(1),256\\ CNN(3),256\\ CNN(1),1024 \end{array}\right]\times 6$	/	/	/
Conv Layer 5	$\left[\begin{array}{c} CNN(1),512\\ CNN(3),512\\ CNN(1),2048 \end{array}\right]\times 3$	/	/	/
Patch Size	/	(4 × 20)	/	/
Head Number	/	12	/	/
Encoder Layer Number	/	8	/	/
Embedding Dropout	/	0.1	/	/
Dropout	/	0.1	/	/
Kernel Size	/	/	2	/
Linear Layer Size	/	/	/	$\left[\begin{array}{c} FC(1024)\\ FC(2048)\\ FC(8) \end{array}\right]$
Dropout	/	/	/	0.2

Note: CNN (a),(b) are convolutional layers, where (a) is the size of convolutional kernel, (b) is the kernel number; FC (d) is fully connected layer, where (d) is the size of FC layer.

**Table 5 sensors-22-06453-t005:** Performance of different sub-module combinations.

Sub-Modules	Metrics	Still	Walk	Run	Bike	Car	Bus	Railway	Subway
R50 (No MAGN)	Accuracy/%	62.93	90.88	98.57	79.50	78.56	70.94	59.76	53.26
	Recall/%	66.34	90.45	98.40	85.46	83.13	65.86	59.02	48.00
	F1-score/%	64.59	90.66	98.49	82.37	80.78	68.31	59.38	50.49
CWT + R50 (No MAGN)	Accuracy/%	56.90	92.85	99.16	93.47	89.49	82.16	68.56	63.60
	Recall/%	80.14	90.28	97.79	93.04	91.35	73.33	58.45	55.35
	F1-score/%	66.55	91.55	98.47	93.25	90.41	77.49	63.10	59.19
CWT + R50	Accuracy/%	78.01	95.89	99.58	91.34	96.99	81.52	73.49	65.26
	Recall/%	88.85	86.77	75.07	94.26	90.42	92.66	79.71	66.52
	F1-score/%	83.08	91.10	85.61	92.78	93.59	86.73	76.48	65.88
CWT + R50 + ECA-net	Accuracy/%	86.87	96.83	98.54	95.15	94.94	94.48	80.69	83.35
	Recall/%	88.73	90.72	99.10	95.30	96.81	92.99	84.72	81.77
	F1-score/%	87.79	93.67	98.82	95.22	95.87	93.73	82.66	82.55
CWT + Vit	Accuracy/%	70.28	86.53	97.73	90.38	88.90	82.34	71.28	63.23
	Recall/%	89.88	86.61	97.79	86.13	85.60	78.34	55.60	67.81
	F1-score/%	78.88	86.57	97.76	88.21	87.22	80.29	62.47	65.44
CWT + R50 + Vit + ECA-net (Proposed Method)	Accuracy/%	88.10	95.34	99.24	96.62	96.49	95.13	88.80	84.72
	Recall/%	91.30	93.68	98.96	95.91	97.34	94.41	85.10	87.42
	F1-score/%	89.67	94.50	99.11	96.26	96.91	94.77	86.91	86.05

**Table 6 sensors-22-06453-t006:** Parameter setting of the baseline methods.

Baselines	Conv Layer	LSTM Layer	Attention Layer	Output Layer
LSTM	/	[LSTM(128)]×1	/	[FC(128)]×1
ABLSTM	/	[LSTM(128)]×1	FC(128)×1 tanh SoftMax	[FC(128)]×1
Deep-Conv LSTM	[CNN(5),64]×4	[LSTM(128)]×2	FC(128)×2 Sigmoid	[FC(128)]×1
TPN	$\left[\begin{array}{c} CNN(24),32\\ CNN(16),64\\ CNN(8),96 \end{array}\right]$ Dropout = 0.1 Maxpool(8)	/	/	[FC(96)]×1
MSRLSTM	$\left[\begin{array}{c} CNN(3),64\\ CNN(2),128\\ CNN(2),128 \end{array}\right]$ Res Maxpool(2)	[LSTM(128)]×2	FC(128)×2 SoftMax	$\left[\begin{array}{c} FC(256)\\ FC(512)\\ FC(1024) \end{array}\right]$

Note: CNN (a),(b) is convolutional layer, where a is the size of convolutional kernel, (b) is the kernel number; LSTM (c) is LSTM layer, where c is the size of hidden layer; FC (d) is fully connected layer, where d is the size of FC layer. Maxpool (e) is the max pooling layer, e is the size of pooling kernel; Res means a residual option after the previous convolutional layer.

**Table 7 sensors-22-06453-t007:** Performance of different baselines and the proposed method.

Baselines	Metrics	Still	Walk	Run	Bike	Car	Bus	Railway	Subway
LSTM	Accuracy/%	56.95	59.98	79.88	50.40	52.18	49.34	35.18	38.22
	Recall/%	60.50	56.86	83.14	56.33	55.18	43.63	28.12	41.60
	F1-score/%	58.67	58.37	81.47	53.20	53.64	46.31	31.25	39.84
ABLSTM	Accuracy/%	50.30	86.43	96.74	62.75	52.90	43.77	41.58	37.52
	Recall/%	77.01	73.06	96.04	60.37	52.34	29.47	33.43	45.13
	F1-score/%	60.86	79.19	96.39	61.54	52.62	35.22	37.06	40.97
Deep-ConvLSTM	Accuracy/%	78.14	92.71	99.38	88.03	87.72	77.66	67.55	61.52
	Recall/%	80.22	87.04	98.03	90.42	86.48	74.27	63.66	69.95
	F1-score/%	79.17	89.79	98.70	89.21	87.10	75.93	65.55	65.46
TPN	Accuracy/%	81.43	99.06	98.59	86.87	67.79	76.08	71.74	56.90
	Recall/%	74.34	61.26	97.35	84.58	95.83	78.16	59.65	70.39
	F1-score/%	77.72	75.70	97.97	85.71	79.40	77.10	65.14	62.93
MSRLSTM	Accuracy/%	75.26	91.94	99.37	90.27	88.34	82.24	71.89	68.36
	Recall/%	86.44	89.69	97.69	91.42	88.53	78.16	65.64	69.23
	F1-score/%	80.47	90.80	98.52	90.84	88.44	80.15	68.57	68.80
Proposed Method	Accuracy/%	88.10	95.34	99.24	96.62	96.49	95.13	88.80	84.72
	Recall/%	91.30	93.68	98.96	95.91	97.34	94.41	85.10	87.42
	F1-score/%	89.67	94.50	99.11	96.26	96.91	94.77	86.91	86.05

## Data Availability

The data is publicly available in reference [5], which can be obtained from: http://www.shl-dataset.org/download/ (accessed on 28 June 2022).

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
