# Peer review of "Transportation Mode Detection Combining CNN and Vision Transformer with Sensors Recalibration Using Smartphone Built-In Sensors"

_sensors, 2022, doi:10.3390/s22176453_

Round 1

Reviewer 1 Report

Glad to review the manuscript (ID: sensors-1815858-peer-review-v1). An approach of transportation mode detection-based smartphone sensors is proposed. Generally speaking, the origination is well built and the idea is interesting. However, I have two concerns to propose, one is what the natural contributions are or alternatively what the applications are as per the authors’ study; another one is the fuzzy figures. Indeed I am checking and thinking about the output charts but too dim to understand. Therefore, my decision is a major revision. In addition, I have some comments to help the authors improve further.

(1) Please check the 5 situations on Page 1 “still, walking, running, vehicles, and rail vehicles”. I prefer it is better to express as “static, walking, running, in-vehicle, and in-railway-train”. Of course, you can think more. In addition, I did not think there is a significant difference between “trains and subways”, the former referring to a tool while the latter is a mode.

(2) Re the literature review, you should not have used this style like “[7] introduce”. In addition, the tense is chaotic, e.g., “[8, 9] used” is past tense and different from “[7] introduce”. It motivates the authors to check more carefully.

(3) The authors should explain what is derived from the noise dataset. How did the authors collect and when? It should be extremely important because deep learning heavily depends on the database. In addition, why is only the saying “hips” a plural noun?

(4) Checking Figure 2, the unit of the x-axis is “seconds”, right? If so, it is too short to express the process clearly. More importantly, Figure 2 was never mentioned or explained in the context.

(5) I am thinking about what the authors explained on Page 11, which is too superficial. You should not repeat again based on Table but rather dig out the reasons why they exhibit such. Please explain the features of different solutions according to the final results. In addition, the item “R50” is not explained.

(6) The conclusion is too rushed to get the core points of this study. I did know what the important/significant findings are.

(7)Last but not least, the language issue is the concern for readability. Understanding authors as not native speakers, I still suggest authors should improve their level.  

Reviewer 2 Report

Comments are attached. 

Reviewer 3 Report

Paper sensors-1815858 “Transportation Mode Detection Combining CNN and Vision Transformer with Sensors Recalibration Using Smartphone Built-in Sensors”

Comments

This study focuses on transportation mode detection combining CNN and vision transformer with sensors recalibration using smartphone built-in sensors. I think the paper fits well the scope of the journal and addresses an important subject. However, a number of revisions are required before the paper can be considered for publication. There are some weak points that have to be strengthened. Below please find more specific comments:

*Abstract: The abstract could be expanded a bit. I particular, I suggest adding a couple of sentences highlighting the contributions of this work to the state of the art and the major outcomes from the experiments.

*Keywords: I suggest adding one or two keywords more.

*Introduction: The introduction section seems to be reasonable. No comments.

*Literature review: The literature review seems kind of short. Please check for the most recent and relevant studies that have been published over the past 2-3 years. It is essential that the literature review is up to date.

*Before discussing the proposed solution approach, I recommend for the authors to create a general discussion regarding the importance of advanced optimization and artificial intelligence algorithms (e.g., heuristics, metaheuristics) for challenging decision problems. There are many different domains where advanced optimization algorithms have been applied as solution approaches, such as online learning, scheduling, multi-objective optimization, transportation, medicine, data classification, and others (not just the decision problem addressed in this study). The authors should create a discussion that highlights the effectiveness of advanced optimization algorithms in the aforementioned domains. This discussion should be supported by the relevant references, including but not limited to the following:

An online-learning-based evolutionary many-objective algorithm. Information Sciences 2020, 509, pp.1-21.

An Adaptive Polyploid Memetic Algorithm for scheduling trucks at a cross-docking terminal. Information Sciences 2021, 565, pp.390-421.

Ambulance routing in disaster response considering variable patient condition: NSGA-II and MOPSO algorithms. Journal of Industrial & Management Optimization 2022, 18(2), p.1035.

Such a discussion will help improving the quality of the manuscript significantly. After this discussion, it would be logical to specifically focus on the description of the proposed solution approach.

*Please provide more details regarding the input data used throughout the experiments. Some supporting references would be helpful to justify the data selection.

*The manuscript contains quite a lot of figures and tables. Please double check and try to provide a more detailed description of these figures and tables where appropriate to make sure that the future readers will have a reasonable understanding of what these figures represent.

*The conclusions section could be a bit more concise. I suggest clearly summarizing the outcomes of this research and provide future research needs.

Round 2

Reviewer 1 Report

Thanks for the authors' revision. 

In the format of writing reference literature, it is better to use the style: " Somebody presented some certain approach of ...[1]". Indeed, I found my advices are negelected in this revised edition. I cannot judge it is a wrong way to use "[1] presented..." but is not sophisticated. Of course, the authors can stick to your view. However, I still would like to recommend authors to consider it especially after reading more papers. 

I did not agree with authors' explanation: I know subway is metro, as a short distance mode. However, longer one should be rail or railway. In Japan, it can be named as high speed railway, of course, which is commen in the Asia. Thus, I think the readers must be confused with the paralleling modes: subway and train. It is very strange. 

Train is a media of communication, that is a tool. The vehicles of subway and railway are named as trains. How to figure out. Thus, it is a big concern. 

Finally, the level of language was improved much. 

Reviewer 3 Report

The authors took seriously my previous comments and made the required revisions in the manuscript. The quality and presentation of the manuscript have been improved. Therefore, I recommend acceptance.

Author Response

We would like to thank the editor and reviewers for the time and valuable comments on our manuscript.